Insight into the PmrB structures of colistin-resistant Gram-negative bacteria through the multi-template ligand-guided homology modeling and in silico mutagenesis

Anuwongcharoen Nuttapat 1
Phanus-umporn Chuleeporn 1
Chatupheeraphat Chawalit 2
Weakwiweak Kamonlak 1
Kaewsai Noramon 3
Eiamphungporn Warawan warawan.eia@mahidol.ac.th 3
1 Department of Community Medical Technology, Faculty of Medical Technology, Mahidol University , Bangkok , Thailand
2 Center for Research Innovation and Biomedical Informatics, Faculty of Medical Technology, Mahidol University , Nakhon Pathom , Thailand
3 Department of Clinical Microbiology and Applied Technology, Faculty of Medical Technology, Mahidol University , Bangkok , Thailand
García-Contreras Rodolfo
Electronic publication date: 2025 Sep 3
Publication date: 2025
Volume: 13
Electronic Location ID: e19945
Received 2025 Apr 1; Accepted 2025 Jul 28
Copyright: ©2025 Anuwongcharoen et al.
Copyright year: 2025
Copyright holder: Anuwongcharoen et al.
License: This is an open access article distributed under the terms of the Creative Commons Attribution License, which permits unrestricted use, distribution, reproduction and adaptation in any medium and for any purpose provided that it is properly attributed. For attribution, the original author(s), title, publication source (PeerJ) and either DOI or URL of the article must be cited.
License URL: https://creativecommons.org/licenses/by/4.0/

Keywords: PmrB, Colistin resistance, Homology modeling, Molecular docking, Molecular dynamics, Structure-based drug design

Funding: National Research Council of Thailand (NRCT) and Mahidol University N42A650351 Office of the Permanent Secretary, Ministry of Higher Education, Science, Research and Innovation RGNS64-146 Mahidol University (Fundamental Fund: fiscal year 2025 by National Science Research and Innovation Fund (NSRF) This project is funded by the National Research Council of Thailand (NRCT) and Mahidol University (Grant no. N42A650351). The computational resources and software utilized for data analysis in this research are supported by the Office of the Permanent Secretary, Ministry of Higher Education, Science, Research and Innovation (Grant no. RGNS64-146) and Mahidol University (Fundamental Fund: fiscal year 2025 by National Science Research and Innovation Fund (NSRF)). The funders had no role in study design, data collection and analysis, decision to publish, or preparation of the manuscript.

==============================
The increasing prevalence of colistin-resistant Gram-negative bacteria necessitates the development of novel therapeutic strategies. PmrB, a histidine kinase involved in colistin resistance, represents a promising drug target. However, the absence of experimentally resolved PmrB structures limits structure-based drug design efforts. This study employed a multi-template, ligand-guided homology modeling approach to construct full-length PmrB models for four pathogens: Acinetobacter baumannii, Escherichia coli, Klebsiella pneumoniae, and Pseudomonas aeruginosa. The resulting models demonstrated high structural integrity, with over 95% of residues located in favored regions and QMEANDisCo global scores ranging from 0.55 to 0.57. Molecular docking simulations guided the selection of representative ligand-bound states for adenosine triphosphate (ATP)-binding site prediction and yielded superior docking scores compared to models generated by AlphaFold, I-TASSER, and SWISS-MODEL. Molecular dynamics (MD) simulations and Molecular Mechanics Generalized Born Surface Area (MM/GBSA) analysis confirmed the stability and binding affinity of the PmrB–ATP complexes, with ΔG values ranging from −27.72 to −47.71 kcal/mol. In silico mutagenesis revealed that the T246A and L344P mutations in K. pneumoniae enhanced ATP binding affinity and protein stability, potentially contributing to colistin resistance. Analysis of the PmrB–ATP complexes identified both conserved and species-specific interactions. This research provides valuable structural models and mechanistic insights into PmrB, supporting future structure-based drug design and the development of novel interventions against colistin-resistant infections.

Introduction

Multidrug-resistant (MDR) Gram-negative bacteria infections are becoming a global problem due to the scarcity of effective antibiotics. The World Health Organization (WHO) has now warned that the world is “running out of antibiotics”, raising concerns that antibiotic resistance is reaching unprecedented levels. The situation has urged the need for alternative antimicrobial drug strategies (Blair et al., 2015; El-Sayed Ahmed et al., 2020; Zhu, Huang & Yang, 2022). Currently, colistin, a potent antibiotic, is the last defense against MDR Gram-negative infections. Its effectiveness lies in its ability to bind to lipid A on the bacterial lipopolysaccharide (LPS), disrupting the outer membrane and leading to bacterial death (Sharma et al., 2022). The widespread use has prompted a rise in colistin-resistant pathogens, presenting a significant healthcare challenge from the limited availability of alternative treatments.

Colistin resistance often arises from chromosomal mutations that are intrinsically non-transmissible through horizontal gene transfer, with the notable exception of the plasmid-mediated mcr gene, which is transferrable (Olaitan, Morand & Rolain, 2014; Blair et al., 2015; Lee et al., 2015; Liu et al., 2016; Poirel, Jayol & Nordmann, 2017). The mechanisms primarily involve alterations to lipid A, the component of LPS targeted by colistin. Despite varying across bacterial species, most resistance mechanisms share a pathway involving modifications to lipid A by adding 4-amino-4-deoxy-L-arabinose (L-ara4N) and/or phosphoethanolamine (PEtN) (Hamel, Rolain & Baron, 2021). These modifications impart a net positive charge to LPS, diminishing its affinity for colistin and thereby conferring resistance.

In particular, several Gram-negative bacteria such as Acinetobacter baumannii, Escherichia coli, Klebsiella pneumoniae, and Pseudomonas aeruginosa commonly develop resistance through changes in the PmrAB two-component system (TCS) (Adams et al., 2009; Aurélie et al., 2014; Lee & Ko, 2014; Quesada et al., 2015). The PmrAB TCS comprises a response regulator (RR)/PmrA and a sensor histidine kinase (HK)/PmrB, enabling signal transduction through protein phosphorylation cycles. The colistin resistance caused by PmrAB TCS is triggered by environmental stimuli or specific genetic mutations, leading to the excessive production of genes that alter the LPS, including the pmrHFIJKLM (arnBCADTEF) operon and the pmrC (eptA) gene (Raetz et al., 2007; Gogry et al., 2021). This ultimately confers resistance to colistin. Noteworthy, the TCSs are generally absent in humans but prevalent in bacteria, thus PmrAB TCS is a promising target for drug development to tackle colistin resistance (Stock, Robinson & Goudreau, 2000).

PmrB is considered a high-potential target for the development of novel antibacterial drugs because of its important roles associated with antibiotic resistance, and virulence in Gram-negative bacteria (Chen & Groisman, 2013). The PmrB is a transmembrane protein consisting of an extracytosolic sensor domain, two transmembrane helices (TM1/TM2) regions, a HAMP domain, a DHp domain, and a catalytic (CA) domain. The sensor domain detects environmental signals, while the transmembrane helices anchor the protein in the membrane. The HAMP domain connects the transmembrane and cytoplasmic regions, and the DHp domain, with a conserved histidine, is crucial for dimerization and autophosphorylation. At the C-terminus, the CA domain binds ATP and transfers the phosphoryl group to the histidine in the DHp domain (Wang, 2012). This intricate structure enables PmrB to sense external stimuli and initiate the downstream cascade of intracellular responses through phosphorylation events, which ultimately regulates gene expression in response to environmental changes. Although the functions of the protein components have been identified, the lack of experimental-derived structural information of the entire PmrB hinders a complete understanding of its molecular mechanism and structure-based virtual screening. Therefore, an advanced homology modeling approach has been employed in this study to predict the spatial arrangement of critical components and enhance our understanding of the protein’s functional sites, including the ATP binding site, of the PmrB models.

Ligand-guided or ligand-steered homology modeling is a refined approach to protein structure prediction, integrating information derived from known ligands to enhance the accuracy of homology models, particularly within the ligand-binding site. This technique leverages the structural constraints imposed by ligand interactions to guide the modeling process, resulting in more functionally relevant to the models, especially in the absence of experimental structural data. This approach is particularly beneficial when modeling proteins with highly flexible or poorly conserved binding sites, where traditional homology modeling may struggle to generate reliable predictions (Phatak, Gatica & Cavasotto, 2010).

While ligand guidance refines the accuracy of the binding site, the overall quality of any homology model is fundamentally dependent on the availability and completeness of its structural templates. This has been a challenge for many transmembrane proteins like PmrB, which often lack full-length experimental structures (Carpenter et al., 2008). The advent of deep learning methods, particularly the AlphaFold Protein Structure Database (Varadi et al., 2022), has revolutionized this landscape by providing highly accurate models. However, recent comparative studies have shown that traditional template-based methods can still produce more accurate models when high-quality, homologous templates already exist (Lee, Su & Tseng, 2022). Conversely, these studies also demonstrate that AlphaFold significantly outperforms template-based methods in cases where no suitable templates are available, although its accuracy may be lower in flexible loop regions (Lee, Su & Tseng, 2022). Therefore, a powerful modern strategy is to employ a multi-template approach that incorporates both experimentally resolved structures and high-quality AI-predicted models, leveraging the strengths of each to generate more complete and reliable protein structures.

Our study aimed to develop reliable PmrB models of multiple Gram-negative pathogens, encompassing A. baumannii, E. coli, K. pneumoniae, and P. aeruginosa. Multi-template ligand-guided homology modeling was used to create the models, and their structural integrity was validated through rigorous testing. In addition, to assess the functional relevance of these models, we employed in silico mutagenesis to investigate the effects of amino acid substitutions reported in the literature to cause colistin resistance. Thus, the set of promising PmrB models was highlighted for further structure-based virtual screening, aiding in discovering new inhibitors to combat antibiotic-resistant strains. Insights from this research may help address antibiotic-resistant strains and develop effective treatment strategies for various Gram-negative pathogens.

Materials & Methods

The research methodology used in this study involved a multi-step process in creating reliable homology models of the PmrB histidine kinase from A. baumannii, E. coli, K. pneumoniae, and P. aeruginosa for use in structure-based virtual screening. The workflow began with retrieving the amino acid sequence from the GenBank database (Sayers et al., 2020) and template identification through Position-Specific Iterated BLAST (PSI-BLAST). Subsequent steps involve multi-template homology modeling, molecular docking for model selection, and structural validation. Furthermore, a structural analysis was undertaken to gain insight into the underlying effects of the PmrB mutation upon binding. The overall process is summarized in Fig. 1.

Figure 1 The schematic workflow of the research methodology used in this study.

Retrieval of amino acid sequence and template identification

The amino acid sequence of the PmrB histidine kinase originating from the selected Gram-negative pathogens was extracted from the GenBank database (Sayers et al., 2020). A comprehensive search was conducted using specified keywords for bacterial strain in combination with “histidine kinase” and “PmrB”. The most relevant sequences (as shown in Table S1) were then selected for downstream analysis.

In order to identify appropriate templates for homology modeling, a Position-Specific Iterated BLAST (PSI-BLAST) search was conducted on the Protein Data Bank (PDB) (Berman et al., 2000). The investigation was performed iteratively twice to minimize incorrect alignments that occurred from randomization. To be considered as templates for homology modeling, each structure must meet specific criteria, including containing the ATPase domain in holo-form with a relevant ligand in the binding pocket, which is essential for maintaining a proper binding site architecture of the protein-ligand complex during model construction. Moreover, we also extended the template searching to the AlphaFold database (Varadi et al., 2022) for use as complementary templates to enhance the predictive performance of homology modeling. The templates that satisfied these requirements are summarized in Table 1.

Table 1 Summary of templates identified from PSI-BLAST and AlphaFold database.

Organisms	Source	Template ID	Ligand	% Identity	% Coverage	
Acinetobacter baumannii	Protein Data Bank	4BIW	ANP	27.27	59	
AlphaFold	L9M1H7	–	95.5	100	
Multi-templates	4BIW-L9M1H7	ANP	–	–	
Escherichia coli	Protein Data Bank	5C93	ACP	29.24	60	
AlphaFold	S1NQ42	–	100	100	
Multi-templates	5C93-S1NQ42	ACP	–	–	
Klebsiella pneumoniae	Protein Data Bank	4BIW	ANP	28.02	65	
AlphaFold	A0A0H3GQH3	–	97.37	94	
Multi-templates	4BIW-A0A0H3GQH3	ANP	–	–	
Pseudomonas aeruginosa	Protein Data Bank	4KP4	ANP	30.59	44	
AlphaFold	Q9HV31	–	100	100	
Multi-templates	4KP4-Q9HV31	ANP	–	–	
Notes.

ACP, Phosphomethylphosphonic Acid-Adenylate Ester; ANP, Phosphoaminophosphonic Acid-Adenylate Ester

Multi-template homology modeling with ligand-specific restraints

To address the challenges posed by a low sequence identity of less than 30 percent on average and a limited amino acid sequence coverage of currently available templates, a multi-template homology modeling strategy was employed using MODELLER (version 10.4) (Webb & Sali, 2016). The approach involved the utilization of multiple templates (Table 1) incorporated with ligand information to construct more precise and dependable models. Prior to performing homology modeling, the target-template sequence alignment was carried out using ClustalOmega (Sievers et al., 2011) and prepared as inputs for homology modeling. Consequently, we generated an ensemble of 1,000 homology models with customized parameters by maximizing the structure refinement and optimization to ensure the accuracy of protein models. FoldX (version 5.0) (Delgado et al., 2019) was employed to provide additional optimization by fixing torsional clashes of amino acid side chains. Furthermore, the amino acid sequences were submitted to the SWISS-MODEL (Waterhouse et al., 2018), I-TASSER (Yang & Zhang, 2015), and AlphaFold (Varadi et al., 2022) to generate homology models for comparison.

Molecular docking and model selection

To select the appropriate protein conformation that matches ATP structure, a natural ligand of histidine kinase, molecular docking was conducted using AutodockFR (Ravindranath et al., 2015). This process was assisted by in-house developed Python scripts and the ChemAxon Marvin Suite for drawing and characterizing chemical structures (ChemAxon, 2023). The genetic algorithm parameters were customized as follows:

• Number of Genetic Algorithm evolutions (nb_runs): 20

• Maximum evaluations of the scoring function per Genetic Algorithm run (max_evals): 4,000,000

• Termination of Genetic Algorithm evaluations after consecutive generations with no improvement (no_improve_stop): 5

• Maximum number of generations (max_gens): 10,000,000

• Seed for the random number generator to ensure reproducibility (seed_value): 1.

The models were ranked based on their binding energy, and the top-ranked models were chosen to compare with the model generated from other approaches. The Protein-ligand Interaction Profiler (PLIP) (Adasme et al., 2021) was also used to analyze the interactions between ATP and PmrB obtained from the best-performing model. The results were visualized in PyMOL software (Schrödinger, 2024) for a comprehensive structural analysis. The transmembrane regions were identified using the Phobius web server (Käll, Krogh & Sonnhammer, 2007). The structural validation was then performed to assess the model quality with multiple validation matrices, including Discrete Optimized Protein Energy (DOPE) score (Shen & Sali, 2006), MolProbity (Williams et al., 2018), QMEANDisCo (Studer et al., 2020), and Ramachandran plots using the SWISS Structure Assessment web server (Waterhouse et al., 2024). The detailed Python scripts implementing this multi-template ligand-guided homology modeling workflow are publicly available in our GitHub repository (https://github.com/nuttapat/Modeller-AutodockFR).

Molecular dynamics simulation

Molecular dynamics (MD) simulations were performed using GROMACS version 2024.3 (Abraham et al., 2024). The topology and parameters for the ATP ligand were obtained from the CGenFF webserver (Vanommeslaeghe, Raman & MacKerell Jr, 2012), ensuring compatibility with the CHARMM36 force field (Vanommeslaeghe et al., 2010), employed for all simulations. Each PmrB-ATP complex was carefully prepared for simulation by solvating it in a cubic box with extended simple point charge (SPC/E) water molecules, ensuring a minimum distance of 10 Å between the protein and the box edges. The protonation states of the N- and C-termini of the protein were fixed to match the requirements of the CHARMM36 force field, and counterions were added to maintain the overall charge neutrality of the system.

At the pre-MD simulations step, the system underwent a two-stage equilibration process, consisting of a 100 ps canonical ensemble (number of particles, volume, and temperature, NVT) equilibration at 300 K followed by a 100 ps isothermal-isobaric ensemble (number of particles, pressure, and temperature, NPT) equilibration at 300 K and 1 bar pressure. Subsequently, two independent MD simulations were then conducted for 200 ns at 300 K and one bar pressure with random seed number. The trajectories of all atoms in the system were recorded and analyzed using Python scripts with the Matplotlib library to calculate various parameters, including the root mean square deviation (RMSD) of the protein backbone and the ATP ligand, the root mean square fluctuation (RMSF) of individual residues, the radius of gyration (Rg) of the protein, the solvent accessible surface area (SASA), and the number of hydrogen bonds formed between PmrB and ATP. Furthermore, the average atomic distance between key interacting residues and functional moieties of ATP during 200 ns simulations was calculated to emphasize the dynamic binding behavior at the binding sites. These investigations provided valuable insights into the stability, flexibility, and interactions of the PmrB-ATP complex.

Binding free energy calculation

Binding free energies between PmrB and ATP were estimated using the Molecular Mechanics Generalized Born Surface Area (MM/GBSA) method implemented in the gmx_MMPBSA tool (Valdés-Tresanco et al., 2021). The generalized Born (GB) model was employed to estimate the polar solvation energy efficiently. Per-residue energy decomposition analysis was performed using the default parameters in gmx_MMPBSA to identify key residues contributing to the PmrB-ATP interaction. The information obtained from this analysis is essential for further understanding the main driving force of the complex formation.

Structural analysis of PmrB mutation

In order to investigate the effects of PmrB mutation on ligand binding, in silico mutagenesis was conducted on the PmrB model of K. pneumoniae due to the availability of a large amount of genetic characterization data that have been previously identified in genomic studies of colistin-resistant K. pneumoniae clinical isolates. This was based on prior research indicating amino acid substitutions that lead to an increase in vitro colistin resistance (Elias, Duarte & Perdigão, 2021). The BuidModel function of FoldX was utilized to assign amino acid substitutions with a particular focus on the histidine kinase domain, including T246A, R256G, K281L, L339C, H340I, N341T, R342D, Q343S, L344P, and P346Q, to the PmrB model and evaluated for internal energy contribution. The molecular docking was performed to calculate the binding energy and predict the binding mode of ATP to the mutants. Finally, the structural comparison between wild-type and mutants was carried out in PyMOL.

Results

Ligand-guided homology modeling

Multi-template homology modeling was performed to create PmrB histidine kinase homology models for four bacterial species: A. baumannii, E. coli, K. pneumoniae, and P. aeruginosa. Suitable templates were identified using PSI-BLAST searches and supplemented with structures from AlphaFold, as shown in Table 1. The search for templates revealed that three histidine kinase crystal structures in complex with their ligand (4BIW, 5C93, and 4KP4) were closely related to our query amino acid sequence, based on the highest alignment score (MaxScore). Although there was limited sequence coverage for PmrB in the selected bacterial species, with a percentage of sequence coverage of 65%, 60%, 44%, and 59%, respectively, the amino acid sequence of all selected templates covered the histidine-kinase ATPase domain. This domain is crucial for studying protein-ligand interaction.

Despite the limitations of low sequence identity and coverage, we utilized multi-template homology modeling in MODELLER to improve the accuracy of protein models, especially at the ATP-binding site, which is covered by a highly flexible lid loop. This involved generating 1,000 protein models for each deployed template (both singles and multiple templates) and refining them using FoldX for side-chain optimization. The models were ranked by calculating the DOPE score, a statistical potential that estimates the energy of the protein structure and distinguishes between proper and incorrect folds. Lower DOPE scores indicate more favorable energy states and may represent more accurate models (Khare et al., 2019). It can be noted that the PmrB models generated using the AlphaFold template exhibit lower DOPE scores compared to the PDB templates identified from the PSI-BLAST, as shown in Table 2 and Table S3. Furthermore, the multi-template models also provide a superior DOPE score than those constructed from a single PDB template. This may result from the inaccuracy of modeling missing regions in sequence alignment. Therefore, utilizing multi-template and ensemble protein modeling may improve the structural prediction of proteins with low sequence identity and coverage to the template. These findings demonstrate the potential of using the AlphaFold as a complementary template, which can be beneficial in the modeling of PmrB structures.

Table 2 Summary of the DOPE and docking scores obtained from ligand-guided homology modeling and other approaches.

Organisms	Software (Template)	DOPE score	Docking score	
		Best	Average	SD	Best	Average	SD	
Acinetobacter baumannii	Modeller (4biw)	−29,492.300	−28,478.09	331.58	−10.60	−7.66	0.73	
Modeller (AlphaFold)	−49,111.780	−48,413.15	243.33	−11.40	−8.78	0.88	
Modeller (multi-templates)	−47,392.140	−46,418.45	342.68	−10.30	−7.34	0.84	
AlphaFold	−51,572.359	–	–	−9.90	–	–	
I-TASSER	−47,302.004	–	–	−6.40	–	–	
SWISS-MODEL	−51,446.324	–	–	−7.50	–	–	
Escherichia coli	Modeller (5c93)	−24,762.390	−23,979.48	268.19	−9.90	−7.11	0.72	
Modeller (AlphaFold)	−39,235.260	−38,559.01	225.11	−10.70	−7.83	0.61	
Modeller (multi-templates)	−37,350.860	−36,347.51	485.00	−10.00	−7.11	1.06	
AlphaFold	−41,230.820	–	–	−8.50	–	–	
I-TASSER	−35,634.230	–	–	−5.30	–	–	
SWISS-MODEL	−40,865.141	–	–	−8.00	–	–	
Klebsiella pneumoniae	Modeller (4biw)	−26,319.050	−25,525.23	277.11	−11.20	−8.94	0.85	
Modeller (AlphaFold)	−36,317.590	−35,396.46	237.17	−10.40	−7.91	0.75	
Modeller (multi-templates)	−34,011.350	−32,519.87	445.64	−10.80	−7.93	0.78	
AlphaFold	−37,045.301	–	–	−6.00	–	–	
I-TASSER	−34,066.145	–	–	−5.60	–	–	
SWISS-MODEL	−36,876.742	–	–	−7.70	–	–	
Pseudomonas aeruginosa	Modeller (4kp4)	−28,425.800	−27,425.48	329.77	−12.80	−9.49	0.83	
Modeller (AlphaFold)	−51,417.220	−50,655.42	246.52	−11.50	−9.18	0.72	
Modeller (multi-templates)	−49,049.800	−48,218.38	358.07	−11.00	−7.42	0.93	
AlphaFold	−53,947.773	–	–	−8.70	–	–	
I-TASSER	−39,335.742	–	–	−4.90	–	–	
SWISS-MODEL	−53,722.395	–	–	−8.50	–	–	

Although the DOPE score can be used as a default scoring for structural assessment, it did not provide information on the arrangement of protein structural motifs and binding pockets. To improve the model selection with precise binding pocket architecture, molecular docking was additionally performed to assess the binding of ATP, the natural ligand of histidine kinases, to all generated PmrB models. This served to select the representative protein conformations based on the binding fitness to the endogenous ligand, thus ensuring the correction of the ligand binding site (Schaller et al., 2019). The best docking score of all PmrB models generated by ligand-guided homology modeling with different types of templates for all bacterial species exhibited superior binding affinity as indicated by docking score in the range of −10.00 kcal/mol to −12.80 kcal/mol, while the average docking scores ranged from −7.11 kcal/mol to −9.49 kcal/mol suggested lower binding affinity to the targets, as shown in Table 2. Interestingly, the docking score of ATP against all generated models from the ligand-guided homology modeling also significantly outperformed the models generated by the AlphaFold, I-TASSER, and SWISS-MODEL. These results highlight the synergistic advantage of combining ensemble model generation with molecular docking for model selection, enabling the identification of PmrB conformations that accurately represent the stable, ligand-bound state. This integrated approach demonstrates its superiority over conventional homology modeling methods by consistently yielding models with improved ATP-binding site architecture, as evidenced by the superior docking scores.

Analysis of the top 10 models, ranked by binding energy from the ligand-guided homology modeling, consistently revealed similar ATP binding sites. In contrast, the top 10 models generated from conventional homology models, which were ranked by DOPE score, exhibited substantial deviations in the ATP binding pocket. This discrepancy arises from variations in ATPase domain orientation and, most notably, the conformation of the flexible lid loop region within the binding pocket (Fig. S1). These findings underscore the importance of utilizing a ligand-guided approach for model selection, prioritizing binding energy as a key metric to ensure an accurate representation of the ligand-bound state. The final refined PmrB models and their binding modality with ATP for A. baumannii, E. coli, K. pneumoniae, and P. aeruginosa are illustrated in Figs. 2A–2D.

Figure 2 The top-ranked PmrB models in monomeric form with ATP-binding conformations.

(A) A. baumannii, (B) E. coli, (C) K. pneumoniae, and (D) P. aeruginosa. The blue solid line, as well as the yellow and green dashed lines, represent the hydrogen bonds, electrostatic interaction, and π − π stacking, respectively.

Structural assessment of PmrB models

The structural integrity and reliability of the final PmrB homology models were rigorously evaluated using multiple assessment metrics, as shown in Fig. 3. MolProbity scores for all models ranged from 2.58 to 2.82, indicating good stereochemical quality. Ramachandran plot analysis further confirmed this, with over 95% of residues in favored and allowed regions with minimal outliers. The overall model quality, as determined by QMEANDisCo global scores, demonstrated consistently high values, ranging from 0.55 to 0.57. This data further validates the structural reliability of the models. Importantly, all models exhibited normalized QMEAN scores (Z-scores) less than 1, indicating quality comparable to experimentally determined structures in the Protein Data Bank.

Figure 3 Structural assessment of PmrB models.

(A) A. baumannii, (B) E. coli, (C) K. pneumoniae, and (D) P. aeruginosa using SWISS Structure Assessment tools, including MolProbity, Ramachandran plots, and QMEANDisCo. The PmrB models show the distribution of phi and psi dihedral angles in Ramachandran plots where the favored regions are represented in dark green, allowed regions in light green, and disallowed regions in white. The structures of the predicted PmrB were also highlighted by the Local Distance Difference Test (IDDT) score, with blue indicating lower atomic distance to AlphaFold’s template and red indicating higher interatomic distances.

Additionally, the DOPE scores of our ligand-guided homology models consistently outperformed those generated using SWISS-MODEL and I-TASSER alone, as summarized in Table 2. This underscores the effectiveness of our modeling strategy, especially for low-sequence identity targets like PmrB. This comprehensive assessment provides strong evidence for the accuracy and suitability of our models for further investigations into PmrB structure, function, and interactions with ATP, paving the way for potential therapeutic interventions against colistin resistance.

Analysis of MD trajectory

To assess the stability and dynamic behavior of the constructed PmrB homology models, we conducted two independent all-atom MD simulations for 200 ns. Details of the dynamic behavior of all MD trajectories were summarized and illustrated in Figs. 4A–4F, Figs. S2, and S3. The RMSD of the protein backbone was analyzed to evaluate the structural deviation over time. As shown in Fig. 4A, all four PmrB models reached equilibrium after approximately 125 ns, indicating that the structures stabilized during the simulation. The average RMSD values for A. baumannii, E. coli, K. pneumoniae, and P. aeruginosa were 1.73 ± 0.26 nm, 0.89 ± 0.14 nm, 1.16 ± 0.15 nm, and 1.89 ± 0.37 nm, respectively. Minor fluctuations were observed in the A. baumannii and P. aeruginosa models, suggesting some degree of flexibility in these proteins. The RMSD of the ATP ligand within the ATP-binding site (Fig. 4B) also showed stabilization with some minor fluctuations later, with average RMSD values of 0.23 ± 0.02 nm, 0.24 ± 0.03 nm, 0.21 ± 0.02 nm, and 0.17 ± 0.03 nm for A. baumannii, E. coli, K. pneumoniae, and P. aeruginosa, respectively. This indicates a lower degree of structural deviation of the ligand inside the binding pocket of PmrB, conferring stability of binding from all constructed models.

Figure 4 Structural dynamics and global stability of PmrB homology models obtained from duplicated MD simulations.

(A) Root mean square deviation (RMSD) of the PmrB protein structures for A. baumannii, E. coli, K. pneumoniae, and P. aeruginosa. (B) RMSD of the ATP ligand within the PmrB ATP-binding site. (C) Root mean square fluctuation (RMSF) of PmrB residues, indicating the flexibility of different regions of the protein. (D) Radius of Gyration (Rg) of PmrB, reflecting the compactness of the protein structure over time. (E) Solvent Accessible Surface Area (SASA) of PmrB, showing the protein’s exposed surface area to the solvent. (F) Number of hydrogen bonds formed between PmrB and the ATP ligand during the simulation.

The flexibility of different regions of the PmrB protein was assessed by analyzing the RMSF of individual residues, as shown in Fig. 4C. It can be noted that loop regions and terminal ends exhibited higher RMSF values, indicating greater flexibility during simulation. The average RMSF values for A. baumannii, E. coli, K. pneumoniae, and P. aeruginosa were 0.58 ± 0.17 nm, 0.49 ± 0.20 nm, 0.54 ± 0.36 nm, and 0.91 ± 0.27 nm, respectively. In contrast, the core regions of the protein showed lower RMSF values, suggesting that these regions maintain a relatively rigid and stabilized structure. Interestingly, similar fluctuation patterns were observed amongst the catalytic domain of all four bacterial species, suggesting the conserved structural dynamics behavior of PmrB.

The overall compactness of the PmrB models was evaluated using the radius of gyration (Rg) as shown in Fig. 4D. The average Rg values for A. baumannii, E. coli, K. pneumoniae, and P. aeruginosa were 4.21 ± 0.18 nm, 3.77 ± 0.09 nm, 3.62 ± 0.18 nm, and 4.54 ± 0.30 nm, respectively. The Rg values for all four species remained relatively stable after an initial simulation period, further supporting the structural stability of the models.

The solvent-accessible surface area (SASA) was analyzed to assess the extent of the protein surface exposed to the solvent. The average SASA values for A. baumannii, E. coli, K. pneumoniae, and P. aeruginosa were 279.99 ± 3.59 nm2, 230.23 ± 2.90 nm2, 236.76 ± 9.93 nm2, and 292.02 ± 10.00 nm2, respectively. While some initial fluctuations were observed in Fig. 4E, particularly for P.aeruginosa, the SASA values generally stabilized over time, suggesting that the overall solvent exposure remained relatively consistent.

We also analyzed the number of hydrogen bonds (Hbond) formed between PmrB and the ATP during the simulation, as demonstrated in Fig. 4F. The average numbers of hydrogen bonds for A. baumannii, E. coli, K. pneumoniae, and P. aeruginosa were 5.35 ± 1.23, 4.58 ± 1.30, 8.09 ± 1.16, and 5.82 ± 1.16, respectively. The plots revealed dynamic fluctuations in the number of hydrogen bonds, which is expected due to the dynamic nature of these interactions. It is notable that the Hbond of K. pneumoniae was significantly greater than that of the others, indicating higher stabilization of the complex during MD simulation. The results regarding the average Hbond offered insights into the overall stability of the PmrB-ATP complex.

MM/GBSA and per-residue decomposition analysis

To further investigate the PmrB-ATP interaction, we performed MM/GBSA analysis to calculate the energy change upon binding (ΔG) for each PmrB-ATP complex. The ΔG values, along with their standard deviations (SD) and standard errors of the mean (SEM), are summarized in Table 3, while the details of energy contributions and per-residue energy decomposition from each MD trajectory are presented in Figs. S4 and S5. The results showed that all four PmrB-ATP complexes exhibited negative ΔG values, indicating that the formation of the complex is energetically favorable from an enthalpic perspective. The individual energy components contributing to ΔG, namely van der Waals (VDWAALS), electrostatic (EEL), polar solvation (EGB), and nonpolar solvation (ESURF) energies, are shown in Figs. 5A–5D. It can be noted that VDWAALS and EEL interactions were the major driving forces for binding in all complexes, contributing significantly to the negative ΔG values. In contrast, the EGB and ESURF terms represented unfavorable contributions due to the desolvation penalty upon complex formation.

Table 3 Energy change upon binding (ΔG) for PmrB-ATP complexes across four bacterial species using MM/GBSA method.

Organisms	ΔG (kcal/mol)	SD (kcal/mol)	SEM (kcal/mol)	
Acinetobacter baumannii	−34.27	7.23	0.05	
Escherichia coli	−27.85	8.11	0.06	
Klebsiella pneumoniae	−47.71	7.29	0.05	
Pseudomonas aeruginosa	−27.72	5.72	0.04	
Notes.

ΔG, average of energy change upon binding; SD, standard deviation; SEM, standard error of mean.

Figure 5 Binding free energy components and per-residue energetic decomposition of MM/GBSA calculation averaged from two MD trajectories of PmrB-ATP interaction.

The left panel (A–D) represents energetic components contributing to the total binding free energy (ΔG), which were calculated using the MM/GBSA method for A. baumannii (A), E. coli (B), K. pneumoniae (C), and P. aeruginosa (D). Energy components include van der Waals (VDWAALS), electrostatic (EEL), polar solvation (EGB), and nonpolar solvation (ESURF) energies. The right panel (E–H) displays per-residue decomposition analysis of the PmrB-ATP binding free energy for A. baumannii (E), E. coli (F), K. pneumoniae (G), and P. aeruginosa (H), highlighting individual residues’ energetic contribution to the PmrB-ATP interaction, where the residues with negative values contribute favorably to the binding.

To identify the key residues involved in PmrB-ATP binding, we conducted per-residue energy decomposition analysis. The results, presented in Figs. 5E–5H, and Table S3, revealed that several amino acid residues made substantial contributions to the ΔG obtained from the PmrB-ATP complex formation. In A. baumannii, GLY399, ASN341, and GLY401 exhibited the most substantial contributions, with average interaction energies contribution of −2.63, and −2.57 kcal/mol, respectively. Similarly, in E. coli, ASN264, TYR268, and LEU322 were the major contributors, with interaction energies ranging from −2.68 to −2.05 kcal/mol. In K. pneumoniae, GLY320, TYR269, GLY322, and ASN265 played a crucial role, with interaction energies contribution ranging from −4.63 to −2.42 kcal/mol. Finally, in P. aeruginosa, ASN363, GLY423, LEU424, and GLY421 were the major contributors, with interaction energies ranging from −6.10 to −2.63 kcal/mol. These residues, along with other key contributors, form a network of interactions that anchor ATP within the binding site. The details of energy contributions and per-residue energy decomposition from each MD trajectory are presented in Figs. S4 and S5.

Interestingly, several residues exhibiting strong favorable interactions with ATP, such as ASN265 (−2.42 kcal/mol) in K. pneumoniae, are not directly involved in binding to the ATP but are located adjacent to the residues in the binding pocket. This suggests that these residues may play a crucial role in supporting the alignment of residues within the binding site, facilitating an induced fit mechanism that enhances the interaction with ATP. This observation underscores the importance of considering not only the direct interactions between the protein and ligand but also the contributions of neighboring residues in mediating the binding process. These findings are consistent with our observation from the in silico mutagenesis study, where mutations in residues adjacent to the binding pocket, such as T246A and L344P in K. pneumoniae, led to significant changes in ATP binding affinity. This further highlights the importance of these neighboring residues in modulating the PmrB-ATP interaction. The structural superimposition between the initial structures and the equilibrated models possessing the lowest ΔG were depicted in Fig. S6.

Structural analysis of PmrB models

The ligand-guided homology models of PmrB across the four Gram-negative pathogens reveal the conserved 3D architecture of this histidine kinase while also highlighting some species-specific variations, as presented in Fig. 2. Each PmrB model consists of a signal peptide, two transmembrane helices (TM1 and TM2), an extracellular sensor domain, a HAMP domain, a DHp domain, and a catalytic domain that includes the ATP-binding site (Wang, 2012). The ATPase lid loop, a flexible region critical for ATP binding, is also located within the catalytic domain. The summary of the amino acid position of each substructural domain from all four bacterial strains was summarized in Table S2.

Across all species, the transmembrane helices (TM1 and TM2) are predicted to be conserved in terms of length and sequence, consistent with their role in anchoring the protein to the cell membrane and mediating signal transduction. In contrast, the extracellular sensor domain, responsible for detecting environmental stimuli, exhibits notable differences in length between the Enterobacterales family (E. coli and K. pneumoniae) and non-fermentative bacteria (P. aeruginosa and A. baumannii). In Enterobacterales, this domain consists of approximately 30 amino acids, whereas in non-fermentative bacteria, it is considerably larger, with 116 and 127 amino acids for A. baumannii and P. aeruginosa, respectively. This variability likely reflects adaptations to the distinct environmental niches and signaling mechanisms employed by these two groups of bacteria.

The HAMP and DHp domains, crucial for signal transduction and autophosphorylation, respectively, are relatively conserved in length and sequence across the four species (Table S2). This conservation underscores their essential role in PmrB function and colistin resistance. The catalytic domain, which contains the ATP-binding site, is the largest domain in all models and shows moderate variations in length. Notably, the ATPase lid loop, a flexible region that regulates access to the ATP-binding site, is highly conserved in length and sequence across all four species, with the exception of P. aeruginosa, which has a slightly longer loop (18 amino acids compared to 16 in the other species). This high degree of conservation suggests a critical role for this loop in ATP binding and PmrB function (Marina et al., 2001; Marina, Waldburger & Hendrickson, 2005; Albanesi et al., 2009; Trajtenberg et al., 2010).

Analysis of protein-ligand interaction profiles of all generated PmrB models in complex with ATP was performed to investigate the key interacting residues and their moieties of preference among PmrB from all four bacterial species. The results reveal a conserved pattern of ATP binding across the four bacterial species, with some notable species-specific variations, as shown in Table S2. All models exhibit hydrogen bonding between the adenine moiety of ATP and residues in the catalytic domain, highlighting the importance of these interactions for ATP binding. Notably, direct π-π stacking interactions were observed between adenine and Y268 in E. coli and Y367 in P. aeruginosa with average atomic distances of 0.25 ± 0.06 and 0.28 ± 0.05 nm, respectively (Fig. 6). In K. pneumoniae and A. baumannii, the corresponding tyrosine residues (Y269 and Y345, respectively) are located close to the adenine moiety, suggesting a strong potential for π − π stacking interactions. Although the distance exceeds the strict criteria for classification by PLIP, the average atomic distances calculated from two independent MD simulations are 0.33 ± 0.12 and 0.30 ± 0.05 nm, respectively, indicating a significant likelihood of binding throughout the simulation. Additionally, all models display hydrogen bonds between the ribose moiety of ATP and residues in the lid loop or catalytic domain, further stabilizing the ligand-protein complex.

Figure 6 Average atomic distance between key interacting residues and functional moieties of PmrB-ATP complexes obtained from MD simulations.

The average atomic distance was derived from duplicated MD trajectories of the PmrB-ATP complex, each lasting 200 ns, for (A) A. baumannii, (B) E. coli, (C) K. pneumoniae, and (D) P. aeruginosa.

While the overall pattern of ATP binding is conserved, species-specific interactions are observed, particularly in the residues involved in hydrogen bonding and salt bridge formation with the phosphate groups of the ATP (Table S4). For example, A. baumannii and E. coli exhibit salt bridges with residues K344 (0.35 ± 0.10 nm) and R310 (0.44 ± 0.18 nm), respectively, while the other species primarily interact with the phosphate groups through hydrogen bonds. Additionally, K. pneumoniae and P. aeruginosa might exhibit hydrogen bonds between the phosphate groups and residues in the DHp domain (T189 and Q193 in K. pneumoniae, Q294 in P. aeruginosa), suggesting a potential role for this domain in stabilizing ATP binding in these species.

The catalytic domain and the lid loop emerge as the primary regions involved in ATP binding across all four species, consistent with their known function in ATPase activity. However, the observed interactions in the DHp domain of K. pneumoniae and P. aeruginosa suggest a more complex interplay between domains in these species. These findings underscore the importance of considering both conserved and species-specific interactions when investigating PmrB function and designing potential inhibitors.

These structural insights, derived from the ligand-guided homology models, provide a foundation for understanding the molecular basis of PmrB function and colistin resistance. The conserved architecture across species suggests a common mechanism of action, while the species-specific variations highlight potential targets for the development of tailored therapeutic interventions. Further investigation of these structural differences may reveal insights into the varying colistin susceptibility observed among different Gram-negative pathogens.

Impact of PmrB mutation upon ATP-binding activity

Despite the major functionality of PmrB being regulated by ATPase activity, there are many reports about mutations located outside the ATPase binding pocket that can increase the level of colistin resistance. In silico mutagenesis of the K. pneumoniae PmrB model was conducted to gain further understanding of the effects of amino acid substitution upon ATP binding. The initial analysis of PmrB models revealed the presence of T246A/L344P double mutations in the original amino acid sequence obtained from the Genbank. Therefore, the wild-type PmrB model was subsequently generated by reverting these mutations to their original residues (A246T, P344L) and will be used as a baseline model for mutational analysis. All mutations that deployed in this study included L17Q, L82R, S85R, T140P, D150H, T157P, S205P, S208N, T246A, R256G, K281L, L339C, H340I, N341T, R342D, Q343S, and P346Q, all of which have been previously associated with colistin resistance. Herein, a schematic summary of all amino acid substitutions and their locations relative to each functional domain on the PmrB structure of K. pneumoniae was depicted in Fig. 7.

Figure 7 Summary of the amino acid substitution site of PmrB mutants from K. pneumoniae.

The orange stick represents the side chain substitution, while the other color represents the amino acid side chain of the wild-type structure.

Interestingly, the results revealed a diverse range of effects on ATP binding affinity and protein stability, as summarized in Table 4 and Table S5. It can be noted that K281L and L339C mutations within the catalytic domain resulted in a decrease in binding affinity of 0.6 and 0.8 kcal/mol, respectively, compared to the wild type. These residues may likely play crucial roles in stabilizing the ATP-bound conformation, and their substitution may disrupt key interactions within the binding pocket. Conversely, the T246A and Q343S mutation in the catalytic domain led to an increase in ATP binding affinity of −1.0 and −0.8 kcal/mol, respectively. The major notable increase in the docking score was observed in T246A and L344P double mutations presented in our constructed PmrB models, which increase the binding affinity of −1.7 kcal/mol, suggesting the synergistic effects to enhance the ATP-binding.

Table 4 The effects of PmrB mutation on protein stability (total energy change) and ATP-binding activity against PmrB of K. pneumoniae using molecular docking.

Mutation	Location	Total energy change (kcal/mol)	Docking score (kcal/mol)	Score difference (kcal/mol)	
L17Q	TM1	−0.073	−9.3	−0.2	
L82R	TM2	0.509	−8.4	0.7	
S85R	TM2	−0.738	−9.3	−0.2	
T140P	HAMP	4.177	−9.3	−0.2	
D150H	DHp	−0.384	−9.5	−0.4	
T157P	DHp	−1.113	−9.4	−0.3	
S205P	DHp	1.329	−9.9	−0.8	
S208N	CA	−0.859	−9.3	−0.2	
T246A	CA	−0.934	−10.1	−1.0	
R256G	CA	1.983	−9.0	0.1	
K281L	CA	0.068	−8.5	0.6	
L339C	CA	2.209	−8.3	0.8	
H340I	CA	0.859	−8.7	0.4	
N341T	CA	0.312	−9.5	−0.4	
R342D	CA	2.989	−8.5	0.6	
Q343S	CA	−0.632	−9.9	−0.8	
L344P	CA	−1.381	−9.8	−0.7	
P346Q	CA	0.330	−9.4	−0.3	
T246A, L344P†	CA	−4.300	−10.8	−1.7	
WT	Wild Type	0.000	−9.1	0.0	
Notes.

TM1, transmembrane region 1; TM2, transmembrane region 2; HAMP, Histidine kinases, Adenylyl cyclases, Methyl-accepting chemotaxis proteins, and Phosphatases domain; DHp, dimerization and histidine phosphotransfer domain; CA, catalytic domain including ATP-binding site.

† This model was constructed by ligand-guided homology modeling based on the PmrB sequence obtained from Genbank (accession number: WP_004886129.1).

In terms of protein stability, mutations T140P in the HAMP domain, R342D in the catalytic domain, S205P in the DHp domain, as well as L82R in the transmembrane region exhibited the most significant destabilizing effects over other mutations in their corresponding domain, with increases in total energy of 4.177, 2.989, 1.329, and 0.509 kcal/mol, respectively, compared to the wild-type baseline. The notable increase of contributed energy in T140P mutation was found in Van der Waals clashes, Backbone HBond, and Sidechain HBond with values of 4.133, 1.873, and 0.589 kcal/mol, respectively, as shown in Table S5. A similar pattern was found in the S205P mutation with the value of 2.560, 1.354, and 0.632 kcal/mol, respectively. The results suggested that these mutations were likely to increase inter-atomic forces and disrupt essential intramolecular interactions, such as hydrogen bonds or salt bridges, leading to a less stable protein than the wild type. In the meanwhile, the total energy change of R342D mutation was driven by Backbone HBond, Sidechain Hbond, Van der Waals, Electrostatics, Solvation polar, and Solvation hydrophobic with the value of 0.451, 0.643, 1.062, 1.6, 0.153, and 0.957 kcal/mol, respectively. Given that the values for Van der Waals, Electrostatics, and Solvation Hydrophobic are relatively high, these interactions are likely the main contributors to the destabilization of the protein upon mutation. A different energy pattern was observed in L82R, which functions to maintain the structural integrity inside the cell membrane. The total energy of this mutation was contributed by Backbone HBond, Electrostatics, Solvation polar, Solvation hydrophobic, Van der Waals clashes, Entropy sidechain, Entropy mainchain, and Torsional clash with the values of 0.02, 0.05, 0.079, 0.121, 0.009, 0.096, 0.122, and 0.064 kcal/mol, respectively. However, these values are all relatively small, so the degree of destabilization in transmembrane mutation may not be very significant as compared to the aforementioned mutations.

The mutations in L344P and T246A in the catalytic domain, T157P in the DHp domain, and S85R in the transmembrane region conferred increased stability. This was shown by a reduction in total energy of −1.381, −0.934, −1.113, and −0.738 kcal/mol, respectively, compared to the wild type. The L344P mutation displays a notable increase in stability, primarily attributed to a substantial favorable change in mainchain entropy, reduced backbone, and torsional clash with the values of −0.955, −0.571, and −0.222 kcal/mol. In contrast, the results of T246A mutation exhibited different energetic changing patterns in which Van der Waals clashes, solvation polar, and entropy sidechain gain more flavor with the values of −1.473, −1.232, and −0.551 kcal/mol, respectively. The DHp domain mutant of T157P achieved more stability, predominantly due to favorable changes in the entropy of the main chain, reduction of backbone clash, and more flavor solvation hydrophobic with the value of −1.007, −0.957, and −0.735 kcal/mol, respectively. Another mutation in the transmembrane region was observed in S85R, which increases stability slightly. The favorable increase of total energy change of this mutant was contributed by a reduction in solvation hydrophobic, Van der Waals, and backbone HBond with the values of −0.826, −0.503, and −0.407 kcal/mol, respectively.

Interestingly, the T246A/L344P double mutant, representing our PmrB models from the obtained K. pneumoniae sequence, presents an intriguing case due to a greater decrease in total energy of −4.3 kcal/mol, which indicates a potential stabilizing effect over other mutants. The favorable energy change of this combination was primarily driven by a significant reduction in torsional clash (−3.717 kcal/mol) and solvation polar (−1.897 kcal/mol), suggesting the positive effects of mutation towards the torsion angles in the protein backbone and interaction between polar groups in the protein against polar solvent.

The results of in silico mutagenesis revealed a diverse range of effects on protein stability and their energy contribution. However, this experiment did not provide a direct explanation of their effects on the ATP binding affinity. Thus, we conducted a molecular docking study between all PmrB mutants with ATP to investigate the effects of this substitution upon binding. The docking scores of all mutants and wild-type PmrB models were summarized in Table 4. It can be noted that the best docking score was observed in T246A/L344P double mutation with the value of −10.8 kcal/mol followed by −10.1, −9.9, −9.9, and −9.8 kcal/mol of T246A, Q343S, S205P, and L344P, respectively. While the wild-type model exhibited −9.1 kcal/mol as a reference point, all aforementioned mutants achieved a significant increase in ATP binding affinity with over 0.5 kcal/mol from the wild type. These findings suggested that the mutation in these positions may confer the intricate relationship between PmrB structure and their ATPase function, which could potentially lead to PmrB signaling and the response to colistin resistance.

Discussion

The absence of experimentally determined PmrB structures has significantly hindered structure-based drug design efforts and highlighted the need for accurate and reliable structural models. However, the development of accurate PmrB structure was challenged by limited sequence identity and coverage to the available homologous template in the Protein Data Bank and presents a significant challenge in structure-based drug design. This problem was observed in the modeling of several transmembrane proteins in which only a few structures were experimentally resolved (Carpenter et al., 2008; Peng et al., 2024). Traditional homology modeling approaches often struggle to generate reliable models in such scenarios, particularly when dealing with flexible regions like the ATPase lid loop, which is crucial for the ligand binding process. This study demonstrates the power of a multi-template ligand-guided homology modeling strategy to overcome these limitations (Phatak, Gatica & Cavasotto, 2010) and constructs high-quality PmrB models suitable for further investigations.

The foundation of our success lies in the integration of multiple templates, including both experimentally determined structures and AlphaFold models, which significantly enhanced the accuracy of the generated models. This approach effectively addressed the challenges posed by low sequence identity and coverage, enabling the prediction of structurally diverse regions and missing segments in the PmrB sequence. Furthermore, the utilization of AlphaFold models as complementary templates proved instrumental in overcoming the limitations posed by low sequence coverage in top-ranked templates obtained from PSI-BLAST. Deployment of AlphaFold as another template enables the homology modeling to predict full-length protein structures, even in the absence of homologous templates (Varadi et al., 2022), and provides the opportunity to generate complete monomeric PmrB models, including regions not covered by experimental structures.

The ligand-guided model selection process by utilizing molecular docking with ATP, ensured the identification of models with the most functionally relevant binding site conformations. This approach surpassed traditional model selection methods based solely on structural metrics, resulting in the obtaining of PmrB models with accurate ATP-binding site architecture. The importance of this ligand-guided approach is further highlighted by the observation that AlphaFold, while excelling at predicting protein structures, may not accurately capture the states of ligand-bound conformations. The higher docking scores (less binding affinity) were observed for AlphaFold models compared to our ligand-guided models. This result suggested that AlphaFold tends to generate closed conformations of the binding site, potentially hindering ligand binding. This highlights the necessity of employing ligand-guided approaches to refine homology models and ensure accurate representation of the ligand-bound state, which is a critical aspect for structure-based virtual screening (Guterres et al., 2021).

Furthermore, the strategic combination of ligand-guided model selection and the integration of multiple templates, encompassing both experimentally resolved crystal structures and AlphaFold models, significantly enhances the overall accuracy of modeling protein structures with low sequence identity and coverage. This integrated approach effectively addresses the challenges associated with modeling such proteins, offering a robust solution for generating reliable structural models. Notably, utilizing molecular docking to guide model selection effectively tackles the problem of the highly flexible lid loop at the entrance of the protein’s binding site. This inherent flexibility often poses difficulties in accurately predicting the ligand-bound conformation, but our ligand-guided approach successfully navigates this challenge, resulting in models that faithfully represent the open, ligand-accessible state of the binding site. This represents a key advantage of our methodology, where accurate representation of the binding site is paramount for identifying potential inhibitors.

To assess the stability and dynamic behavior of the constructed PmrB homology models, we conducted 200 ns MD simulations in duplicates with random seed numbers for each complex. Analysis of the MD trajectories revealed that the models were stable and maintained their structural integrity throughout the simulation, lending further support to the reliability of our homology modeling approach. The observation that the protein structure and the ATP ligand reached equilibrium relatively quickly at the initial of simulation suggests that the models generated by our ligand-guided approach achieve energetically favorable conformations. The flexibility observed in the loop regions and terminal ends, while maintaining a stable core structure in DHp and catalytic domain, is consistent with the expected behavior of proteins, further validating the models’ accuracy. The stable Rg and SASA values indicate that the overall shape and solvent exposure of the PmrB models remain consistent, reinforcing their structural integrity. Furthermore, the analysis of hydrogen bond interactions revealed stable binding between PmrB and ATP. These findings not only validate our homology models but also provide valuable insights into the dynamic behavior of PmrB, which is crucial for modulating the protein’s function.

The MM/GBSA analysis provided a quantitative assessment of the PmrB-ATP interaction, further strengthening our understanding of the dynamic behavior of these models. It can be noted that the PmrB-ATP complex formation is energetically favorable in all four bacterial species, with K. pneumoniae exhibiting the strongest enthalpic interactions driven by substantial van der Waals and electrostatic contributions. Interestingly, the per-residue energy decomposition analysis provided a more nuanced understanding of the binding event. It showed that some of the residues with strong favorable interactions with ATP, such as ASN261, ARG311, and ASP313 in K. pneumoniae, are not directly involved in binding but are located adjacent to the active site. This suggests these residues play a crucial role in supporting the alignment of the binding pocket, likely facilitating an induced-fit mechanism that enhances the interaction with ATP. This observation underscores the importance of considering the contributions of the wider binding region, not only the direct protein-ligand contacts.

The significance of these adjacent residues is further reinforced by our in silico mutagenesis study, where mutations in these peripheral positions, such as T246A and L344P in K. pneumoniae, led to significant changes in ATP binding affinity. This consistency between the energy decomposition and mutagenesis results highlights the importance of these residues in modulating the PmrB-ATP interaction. Therefore, the ability of the ligand-guided modeling approach to accurately capture the architecture of this entire binding region provides a more reliable structural basis for future drug design. These findings offer valuable insights for further investigation, suggesting that targeting these secondary residues could be a viable strategy for inhibitor development.

The in silico mutagenesis analysis was performed to probe the functional implications of clinically relevant PmrB mutations in K. pneumoniae, thereby bridging the gap between our structural models and the mechanisms of colistin resistance. The findings suggest that resistance-conferring mutations in PmrB do not follow a single mechanistic route but can be broadly categorized into two groups: those that appear to enhance protein stability and ATP affinity, and those predicted to be structurally destabilizing.

A key group of mutations, including T246A and L344P, appears to function by creating a hyperactive state. Our models predict these substitutions enhance protein stability and increase ATP binding affinity, which would possibly accelerate the kinase’s autophosphorylation cycle and upregulate the downstream LPS modification pathway. The molecular basis for these effects can be inferred from the mutational position and their substituted amino acid properties. For the T246A substitution, the replacement of a bulkier threonine with a smaller alanine may relieve local steric clashes, allowing the ATP-binding pocket to settle into a lower-energy, higher-affinity conformation. In the case of the L344P mutation, the introduction of a rigid proline residue into what is likely a flexible loop can lock it into a conformation that is optimal for protein stability and ATP coordination. This is strongly supported by our energy analysis, which showed a significant favorable change in mainchain entropy and a reduction in torsional clash for this mutant. This interpretation is strongly supported by clinical evidence, as these mutations are frequently reported in colistin-resistant K. pneumoniae isolates (Jaidane et al., 2018; Mathur et al., 2018; Pragasam et al., 2017). The predicted synergistic effect of the T246A/L344P double mutant is particularly noteworthy, as the co-occurrence of mutations at these positions has been identified in clinical isolates exhibiting high-level colistin resistance (Pragasam et al., 2017). Our model thus provides a compelling molecular rationale for the resistance observed in these clinical settings.

On the other hand, our analysis predicted that another set of clinically relevant mutations, such as L82R and T140P, would be structurally destabilizing. While it may seem counterintuitive, protein destabilization can also lead to a gain-of-function phenotype by altering the conformational dynamics required for regulation, potentially enabling the kinase in an autophosphorylation state. This hypothesis is supported by existing literature in which the L82R substitution was observed in a patient with an invasive infection and was sufficient to increase the colistin minimum inhibitory concentration (MIC) up to 64-fold higher than the wild type (Cannatelli et al., 2014). Similarly, the T140P mutation, located in the critical HAMP signal transducer domain, has been documented in resistant isolates. However, the presence of this mutation was predicted to be deleterious (Olaitan, Morand & Rolain, 2014). This detrimental effect may result from the introduction of a rigid kink due to proline substitution, which increases the Van der Waals clashes and could hinder the domain’s conformational changes that are essential for signal transduction. Our analysis also revealed that this destabilization mechanism of the mutants can occur through various disruptive effects across different functional domains, which further illustrates that resistance can arise from numerous distinct alterations to PmrB stability and dynamics.

Although our models can provide insight into the structure of PmrB mutations at the molecular level, experimental validation should be conducted to confirm the prediction results. Future studies should employ systematic site-directed mutagenesis to engineer representative mutations from both categories (e.g., the stability-enhancing T246A/L344P and the destabilizing L82R) into a wild-type strain of K. pneumoniae. Subsequent characterization, including colistin MIC determination and in vitro kinase assays to measure ATPase activity, would elucidate the diverse molecular strategies that lead to PmrB-mediated colistin resistance and fully validate the predictions of our structural models.

Ultimately, this work overcomes a significant challenge in the field by delivering the reliable full-length models of PmrB for major Gram-negative pathogens (Qihong et al., 2020; Basu, Veeraraghavan & Anbarasu, 2024). Although our models can provide insight into the structure of PmrB mutations at the molecular level, the mechanistic hypotheses generated underscore the need for experimental validation. Future studies should systematically employ site-directed mutagenesis to engineer representative mutations related to the colistin-resistant K. pneumoniae. Subsequent characterization, including colistin MIC determination and in vitro enzymatic assays to measure ATPase activity, would elucidate the mechanistic insights that lead to PmrB-mediated colistin resistance and fully validate the predictions of our structural models.

Conclusions

This study successfully addressed the critical challenge of generating accurate and complete structural models of PmrB, a key target for combating colistin resistance in Gram-negative bacteria. By employing a multi-template ligand-guided homology modeling approach that integrated both experimentally resolved structures and AlphaFold models, we addressed the challenges posed by low sequence identity and coverage to produce high-quality models with accurate ligand binding site architectures representing the ligand-bound conformation of PmrB. The stability and dynamic behavior of these models were confirmed through various structural assessment metrics, extensive MD simulations, and MM/GBSA analysis. The subsequent structural and mutational analyses provided valuable insights into conserved ATP-binding interactions and the mechanistic impact of clinically relevant resistance mutations. In conclusion, the comprehensive PmrB models generated and assessed in this research represent a valuable resource to guide future structure-based drug discovery efforts against the growing threat of antimicrobial resistance.

Supplemental Information

Supplemental Information 1 A comparison of the top-ranked PmrB models of Klebsiella pneumoniae in monomeric form

(A) Models ranked by binding energy from ligand-guided homology modeling. (B) Models ranked by DOPE score from conventional multi-template homology modeling. The aligned structures are highlighted by their RMSD values, with dark blue indicating low RMSD and red indicating high RMSD compared to the top-ranked model from each approach.

Supplemental Information 2 Analysis of the first molecular dynamics (MD) trajectory of PmrB homology models

(A) Root Mean Square Deviation (RMSD) of the PmrB protein structures for A. baumannii, E. coli, K. pneumoniae, and P. aeruginosa. (B) RMSD of the ATP ligand within the PmrB ATP-binding site. (C) Root Mean Square Fluctuation (RMSF) of PmrB residues, indicating the flexibility of different regions of the protein. (D) Radius of Gyration (Rg) of PmrB, reflecting the compactness of the protein structure over time. (E) Solvent Accessible Surface Area (SASA) of PmrB, showing the protein’s exposed surface area to the solvent. (F) Number of hydrogen bonds formed between PmrB and the ATP ligand during the simulation.

Supplemental Information 3 Analysis of the second molecular dynamics (MD) trajectory of PmrB homology models

(A) Root Mean Square Deviation (RMSD) of the PmrB protein structures for A. baumannii, E. coli, K. pneumoniae, and P. aeruginosa. (B) RMSD of the ATP ligand within the PmrB ATP-binding site. (C) Root Mean Square Fluctuation (RMSF) of PmrB residues, indicating the flexibility of different regions of the protein. (D) Radius of Gyration (Rg) of PmrB, reflecting the compactness of the protein structure over time. (E) Solvent Accessible Surface Area (SASA) of PmrB, showing the protein’s exposed surface area to the solvent. (F) Number of hydrogen bonds formed between PmrB and the ATP ligand during the simulation.

Supplemental Information 4 Analysis of binding free energy components and per-residue decomposition from the first molecular dynamics (MD) trajectory of PmrB-ATP interaction

The left panel (A-D) represents energetic components contributing to the total binding free energy (ΔG), which were calculated using the MMGBSA method for A. baumannii (A), E. coli (B), K. pneumoniae (C), and P. aeruginosa (D). Energy components include van der Waals (VDWAALS), electrostatic (EEL), polar solvation (EGB), and nonpolar solvation (ESURF) energies. The right panel (E-H) displays per-residue decomposition analysis of the PmrB-ATP binding free energy for A. baumannii (E), E. coli (F), K. pneumoniae (G), and P. aeruginosa (H) highlighting individual residues’ energetic contribution to the PmrB-ATP interaction, where the residues with negative values contribute favorably to the binding.

Supplemental Information 5 Analysis of binding free energy components and per-residue decomposition from the second molecular dynamics (MD) trajectory of PmrB-ATP interaction

The left panel (A–D) represents energetic components contributing to the total binding free energy (ΔG), which were calculated using the MMGBSA method for A. baumannii (A), E. coli (B), K. pneumoniae (C), and P. aeruginosa (D). Energy components include van der Waals (VDWAALS), electrostatic (EEL), polar solvation (EGB), and nonpolar solvation (ESURF) energies. The right panel (E-H) displays per-residue decomposition analysis of the PmrB-ATP binding free energy for A. baumannii (E), E. coli (F), K. pneumoniae (G), and P. aeruginosa (H) highlighting individual residues’ energetic contribution to the PmrB-ATP interaction, where the residues with negative values contribute favorably to the binding.

Supplemental Information 6 Comparison of initial and equilibrated PmrB homology models from MD simulations

The equilibrated structure corresponds to the frame with the lowest ΔG after 50 ns equilibration. (A) A. baumannii. (B) E. coli. (C) K. pneumoniae. (D) P. aeruginosa. Left panel: initial structure (green); Middle panel: lowest ΔG structure (white); Right panel: superimposition. This figure demonstrates the stability of the PmrB models and highlights conformational changes upon equilibration, particularly in loop regions and the ATP-binding site. The close agreement between structures supports the reliability of the models for further investigations.

Supplemental Information 7 Amino acid sequences of selected bacterial PmrB for constructing ligand-guided homology models

Supplemental Information 8 The amino acid position of PmrB substructural domains

Supplemental Information 9 Additional data

(A) Calculated DOPE score of all generated models using MODELLER software. (B) The binding energy (affinity) between the ATP and all generated models using AutodockFR. (C) MM/GBSA Binding Energy Terms for PmrB-ATP Complex across Multiple Organisms and Replicate Trajectories. (D) Per-residue energy decomposition analysis of PmrB-ATP interactions from MM/GBSA calculation.

Supplemental Information 10 Protein-ligand interaction profile of all PmrB models in complex with ATP

Supplemental Information 11 Comparison of the free energy changes in PmrB mutants of K. pneumoniae compared to the wild type

The authors of this research article acknowledge the use of AI-powered language editing tools such as Grammarly and Gemini Advanced. The experimental design, computation studies, and all data analysis were conducted solely by the authors without AI assistance. Additionally, the authors manually created all Figures and Tables and were primarily responsible for the preparation and writing of the article. The authors also reviewed and ensured the accuracy of the contents.

Additional Information and Declarations

Competing Interests

Author Contributions

Data Availability

The authors declare there are no competing interests.

Nuttapat Anuwongcharoen conceived and designed the experiments, performed the experiments, analyzed the data, prepared figures and/or tables, authored or reviewed drafts of the article, provided the resource, and approved the final draft.

Chuleeporn Phanus-umporn conceived and designed the experiments, performed the experiments, analyzed the data, prepared figures and/or tables, authored or reviewed drafts of the article, provided the resource, and approved the final draft.

Chawalit Chatupheeraphat conceived and designed the experiments, analyzed the data, authored or reviewed drafts of the article, and approved the final draft.

Kamonlak Weakwiweak performed the experiments, authored or reviewed drafts of the article, and approved the final draft.

Noramon Kaewsai performed the experiments, authored or reviewed drafts of the article, and approved the final draft.

Warawan Eiamphungporn conceived and designed the experiments, authored or reviewed drafts of the article, supervised the study, and approved the final draft.

The following information was supplied regarding data availability:

Code and raw data are available at Github and Zenodo:

https://github.com/nuttapat/Modeller-AutodockFR.

Nuttapat Anuwongcharoen. (2025). nuttapat/Modeller-AutodockFR: Version 1.0 (v1.0.0). Zenodo. https://doi.org/10.5281/zenodo.15943845.

All other raw data, including the final structural models, raw molecular dynamics trajectories, full docking log files, and FoldX outputs, are available at Figshare:

Anuwongcharoen, Nuttapat; Phanus-umporn, Chuleeporn; Chatupheeraphat, Chawalit; Weakwiweak, Kamonlak; Kaewsai, Noramon; Eiamphungporn, Warawan (2025). Supplemental Data - Structure and Simulation Raw Data. figshare. Dataset. https://doi.org/10.6084/m9.figshare.28675244.v2.

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
