# Peer review of "Insight into the PmrB structures of colistin-resistant Gram-negative bacteria through the multi-template ligand-guided homology modeling and in silico mutagenesis"

_PeerJ, doi:10.7717/peerj.19945_

## Round 0.1 · original submission · Minor Revisions

Please address all reviewer comments.

·

Basic reporting

The manuscript by Anuwongcharoen et al has used a multi-template ligand-based homology modelling to develop the structure of the PmrB histidine kinase involved in the development of colistin resistance among the Gram-negative bacterial pathogens. Overall, the work has been performed with clearly defined objectives, the methodologies have been described in detail, and the results have been analysed appropriately with suitable data presentations in the figures and tables.

Experimental design

No comments

Validity of the findings

No comments

Additional comments

The manuscript is written in a lucid manner, which is easy to follow. However, I feel the discussion and the conclusions are repetitive, specifically, how the PmrB model building can aid in the development of drugs has been stated multiple times in the text. This may be looked into.

Reviewer 3 ·

Basic reporting

-

Experimental design

-

Validity of the findings

The authors should use experimental data to validate whether the potential PmrB mutant affects its ability to regulate microbial colistin resistance.

Additional comments

The manuscript shows that a multi-template ligand-guided homology modeling approach can offer a more dependable foundation for protein-ligand interactions. The histidine kinase PmrB is known as a bacterial two-component system that regulates colistin resistance. In this study, the authors employed a multi-template ligand-guided homology modeling method to develop the complete modeling structure of PmrB from four different pathogens. The intricate interactions between the in silico mutated ATP-binding residues and ATP offer structural insights that clarify the effects on ATP binding affinity. Nonetheless, this study did not confirm the effect of specific residues on PmrB in colistin resistance through experimental data. The authors should analyze how known mutated residues associated with colistin resistance affect protein-ligand interactions using the PmrB modeling structure in this study. This validation can demonstrate that the multi-template ligand-guided homology modeling structure yields more reliable insights for future antimicrobial drug development.

Reviewer 4 ·

Basic reporting

1. Language & style.
o English is generally clear, technically accurate, and professional.

2. Background & literature context.
o The Introduction provides a solid overview of colistin resistance, PmrAB TCS biology, and the rationale for modeling PmrB (Blair et al. 2015; Stock et al. 2000).
o Key prior studies on ligand-guided homology modeling are cited (Phatak et al. 2010), but the authors might discuss more recent uses of AlphaFold as a template (Varadi et al. 2022) earlier in the background section.

3. Structure, figures, tables & raw data.
o Follows standard sections (Abstract, Introduction, Methods, Results, Discussion, Conclusions).
o Figures (Figs 1–7, S1–S6) are appropriate, well-labeled, and high resolution.
o Tables 1–4 and Supplementary Tables S1–S5 cover templates, scores, ΔG values, and mutational effects.
o Raw data deposition: The manuscript notes “7 Figure file(s), 5 Table file(s), 10 Other file(s)” but does not explicitly state where raw MD trajectories, docking logs, or FoldX outputs are publicly available. Recommend the authors deposit these data in a recognized repository (e.g., Zenodo, Dryad) and cite accession numbers

Experimental design

1. Original primary research & scope.
o Clearly original computational work in protein modeling and antibiotic resistance; fits PeerJ’s scope.

2. Research question & knowledge gap.
o Well defined: absence of full-length PmrB structures is an obstacle to drug design. The gap is explicit and addressed.

3. Rigor & ethical standards.
o Computational methods follow best practices (duplicate MD runs, multiple validation metrics). No ethical concerns (no human/animal data).

4. Methods detail & reproducibility.
o Thorough descriptions of homology modeling parameters, docking algorithms/settings, MD protocols, and MM/GBSA procedures.
o Suggest providing key scripts or parameter files (e.g., MODELLER Python scripts, GROMACS .mdp files) as supplementary materials or a GitHub link.

Validity of the findings

1. Novelty & replication.
o This is novel structural modeling rather than replication; replication is appropriate only if clearly justified, which is the case for mutagenesis comparisons.

2. Data robustness & controls.
o Docking, MD, and free energy calculations appear well controlled (internal comparisons across different modeling approaches).
o Raw trajectories and energy outputs should be made accessible for community scrutiny.

3. Conclusions & linkage to results.
o Conclusions are well supported by docking scores, MD stability metrics, and MM/GBSA data.

Additional comments

In this work, the authors seek to generate reliable, full-length structural models of the sensor histidine kinase PmrB from four major colistin-resistant Gram-negative pathogens (A. baumannii, E. coli, K. pneumoniae, P. aeruginosa) in order to enable structure-based drug design against colistin resistance. PmrB mediates lipid A modifications that underlie chromosomal colistin resistance; however, no experimental full-length structures exist, hindering inhibitor development. By providing high-quality models of PmrB in its ATP-bound state and exploring mutational effects, the work fills a critical gap toward novel therapeutics.

• Methods.
1. Sequence retrieval & template identification: PmrB sequences were obtained from GenBank, and suitable ATPase-domain templates were identified via iterative PSI-BLAST (from PDB and AlphaFold).

2. Multi-template, ligand-guided homology modeling: MODELLER + ligand restraints to build 1,000 models/species; side-chain optimization with FoldX.

3. Model selection: Molecular docking of ATP with AutoDockFR to rank ligand-bound conformations; comparison against AlphaFold, I-TASSER, and SWISS-Model outputs.

4. Structural validation: DOPE scores, MolProbity, Ramachandran, QMEANDisCo assessments.

5. Molecular dynamics & MM/GBSA: 2×200 ns GROMACS runs per complex; analysis of RMSD, RMSF, Rg, SASA, hydrogen bonds; binding free energies and per-residue decomposition via gmx_MMPBSA.

6. In silico mutagenesis: FoldX-based mutational scanning of known resistance-associated substitutions (e.g., T246A, L344P) in the K. pneumoniae model, followed by docking to assess changes in ATP affinity and protein stability.
• Major conclusions.
o Ligand-guided, multi-template modeling outperforms unguided methods (AlphaFold, I-TASSER, SWISS-Model) in producing PmrB structures with superior ATP-binding site geometry and docking scores (≈–10 to –12.8 kcal/mol).
o MD simulations confirm structural stability (backbone RMSD equilibrating by ~125 ns; consistent Rg/SASA; persistent H-bonds), validating model robustness.
o MM/GBSA shows favorable binding free energies (ΔG ≈ –27.7 to –47.7 kcal/mol), driven by van der Waals and electrostatic terms, with key conserved and species-specific residues identified.
o In silico mutagenesis pinpoints mutations (notably T246A, L344P singly and in combination) that enhance ATP binding affinity and alter protein stability, suggesting mechanistic contributions to colistin resistance.
Overall, this work is novel and solid. I just have a few minor comments to improve the manuscript for publication.
• Data availability: Mandate deposition of raw data (MD trajectories, docking logs, modeling scripts) in a public repository with accession details.
• Supplementary methods: Provide key parameter files and scripts to fully enable reproducibility.
• Enhanced discussion: Briefly consider potential experimental follow-up (e.g., site-directed mutagenesis assays) to validate predicted mutational impacts.

---

## Round 0.2 · accepted · Accept

Thanks for addressing all reviewers' comments!

·

Basic reporting

No comments

Experimental design

No comments

Validity of the findings

No comments

Additional comments

The revised version of the manuscript has addressed the comments made by this reviewer. The revised manuscript is significantly improved in reference to the discussion and removal of the redundancy present previously.

Reviewer 3 ·

Basic reporting

no comment

Experimental design

no comment

Validity of the findings

no comment

Reviewer 4 ·

Basic reporting

The authors have addressed all my comments. The manuscript is good to go.

Experimental design

The authors have addressed all my comments. The manuscript is good to go.

Validity of the findings

The authors have addressed all my comments. The manuscript is good to go.